# Laboratory Experimental Study on the Formation of Authigenic Carbonates Induced by Microbes in Marine Sediments

**Zilin Wei [1], Tianfu Xu [1], Songhua Shang [1], Hailong Tian [1,\*], Yuqing Cao [1], Jiamei Wang [2], Zhenya Shi [3] and Xiao Liu [4]**

[1] Key Laboratory of Groundwater Resources and Environment, Ministry of Education, Jilin University, Changchun 130012, China; zilin_wei@163.com (Z.W.); tianfu_xu@jlu.edu.cn (T.X.); shangsh1995@foxmail.com (S.S.); caoyq_hu@126.com (Y.C.)

[2] College of Construction Engineering, Jilin University, Changchun 130012, China; wangjiamei0203@foxmail.com

[3] Zhongji Yulong Construction Co., Ltd., Nanyang 473000, China; m78kwia@dingtalk.com

[4] The Second Geological Brigade of Shandong Geological Exploration Bureau, Shandong Provincial Lunan Geo-Engineering Exploration Instituts, Yanzhou 272100, China; xiaolkaixin@163.com

\* Correspondence: thl@jlu.edu.cn

**Abstract:** Authigenic carbonates are widely distributed in marine sediments, microbes, and anaerobic oxidation of methane (AOM) play a key role in their formation. The authigenic carbonates in marine sediments have been affected by weathering and diagenesis for a long time, it is difficult to understand their formation process by analyzing the samples collected in situ. A pore water environment with 10 °C, 6 MPa in the marine sediments was built in a bioreactor to study the stages and characteristics of authigenic carbonates formation induced by microbes. In experiments, $FeCO_3$ is formed preferentially, and then $FeCO_3$-$MgCO_3$ complete isomorphous series and a small part of $CaCO_3$ isomorphous mixture are formed. According to this, it is proposed that the formation of authigenic carbonates performed by AOM and related microbes needs to undergo three stages: the rise of alkalinity, the preferential formation of a carbonate mineral, and the formation of carbonate isomorphous series. This work provides experimental experience and reference basis for further understanding the formation mechanism of authigenic carbonates in marine sediments.

**Keywords:** anaerobic oxidation of methane; authigenic carbonates; bioreactor experiment

## 1. Introduction

The marine sediments are the largest methane reservoir on the earth. The methane in marine sediments was stored in the form of free methane, dissolved methane, and natural gas hydrates [1]. The annual methane generated from marine sediments reaches 85–300 Tg [2–4]. However, only 2% of methane generated was emitted into the atmosphere [5]. Previous work estimated that more than 80% of the methane migrating upward through marine sediments was consumed through anaerobic oxidation of methane coupled with sulphate reduction (AOM) mediated by the microbial consortium of anaerobic methane-oxidizing archaea and sulfate-reducing bacteria [6,7]:

$$CH_4 + SO_4^{2-} \rightarrow HCO_3^- + HS^- + H_2O \tag{1}$$

$HCO_3^-$ and $HS^-$ was produced by AOM, which increases the alkalinity of pore water, thus favoring the precipitation of authigenic carbonates and pyrite [8]. Authigenic carbonate was formed by the interaction between bicarbonate, calcium, iron, and magnesium within the pore water of the marine sediment [9]. Authigenic carbonates were very common in several methane-bearing cold seeps where the AOM was occurred. It was formed in different types such as calcite, siderite, aragonite, high-magnesium calcite, low-magnesium calcite [10], dolomite, and ankerite in methane-bearing cold seeps of different areas.

The study of authigenic carbonates in the cold seep area mainly focus on the determination of the origin, formation time of authigenic carbonates, and the source of seepage fluid by isotope method [11–13]. Some papers have discussed the different formation model of authigenic carbonates in the methane-bearing cold seep by studying sediment cores [14,15]. However, the particle size of carbonates in marine sediments is very small, and it is difficult to obtain single component minerals by conventional operation. Therefore, most studies on authigenic carbonates in the cold seep area are limited to determine the total amount of minerals [16], and few studies on the formation process of authigenic carbonate.

The laboratory experiment related to authigenic carbonates, some study the formation duration of carbonate minerals under cold seep conditions [17], and some study the relationship between biominerals and the microbes [18]. However, there are few researches on the formation mechanism of carbonate minerals in marine sediments. The previous research is limited to the exploration of the forming conditions of authigenic carbonates [19] and the simple description of the mineral morphology [20].

In this paper, artificial pore water, marine sediments collected from the South China Sea, and microbial solution incubated from sediments were used to study the formation stage and characteristics of authigenic carbonates induced by microbes in a bioreactor with low temperature and high pressure. The results reflect the stages and model of authigenic carbonates formation in marine sediments to some extent. This work improves our understanding of the formation of authigenic carbonates from a microscopic perspective.

## 2. Materials and Methods

### 2.1. Preparation of Materials

#### 2.1.1. Vitro Incubation of Microbes Strain

The basal medium composition was shown in Appendix A. 10 g seabed sediment from the northern continental slope of the South China Sea mixed 250 mL sterile anaerobic medium was put into culture flask. After 5 min of $N_2$ (99.99% purity, Dalian Special Gas Corporation) flushing and 2 min of $CH_4$ (99.99% purity, Dalian Special Gas Corporation) flushing, culture flask was placed at 35 °C during incubation. To supplement the carbon substrate, excess methane was replenished every two days. Black crusts stuck on the inner wall of culture flask, some black suspension was generated after 10 days' culture. Then, the supernatant of culture flask was obtained and placed it into fresh medium as new enrichment cultures. Repeating this process until the microbial concentration and purity met the requirements (OD value of microbial solution reach 2.1), an enriched microbial solution was obtained.

#### 2.1.2. Artificial Pore Water

The pore water composition, 4.75 m below seafloor in site T1 of Qiongdongnan basin, the South China Sea, was referred to configure artificial pore water (Table 1). At site T1, the ratio of $Mg^{2+}/Ca^{2+}$ rose and the concentration of $Mg^{2+}$, $Ca^{2+}$, $Sr^{2+}$, $Mn^{2+}$, and $SO_4^{2-}$ reduced with depth [21]. These geochemical variations were consistent with the characteristics of pore water variations in the marine sediments [22].

**Table 1.** The composition of artificial pore water.

| Chemical Species | Concentration (mM) | Chemical Species | Concentration (mM) |
| --- | --- | --- | --- |
| $NH_4^+$ | 3.4 | $K^+$ | 15.3 |
| $Mg^{2+}$ | 57.9 | $Ca^{2+}$ | 9 |
| $SO_4^{2-}$ | 22.4 | $Cl^-$ | 603.3 |

#### 2.1.3. Sediments and Granite Slices

Sediment samples taken from the seabed of the Northern continental slope of the South China Sea, were stored at 4 °C in the dark for 6 months until in the experiment started. X-ray diffraction (XRD) and Fourier transform infrared spectroscopy (FTIR) were

performed to analyze the sediments. The result shows that the sediments consist of (in wt.%) quartz 27, plagioclase 3, kaolinite 10, illite 45, and smectite 15 on average.

Limited by the laboratory experiment conditions, the amount of authigenic minerals formed during the experiment was small, which makes it difficult to collect and test. Granite slices, which were washed in adequate dilute hydrochloric acid (3.6 in wt.%) and then rinsed by distilled water and dried at 35 °C before the experiment, were used as intermediate carriers to collect authigenic minerals on it. Before the experiment, the results of XRD show that the granite slices consist of (in wt.%) 22 quartz, 33 plagioclase, 37 alkali feldspar, 5 biotite, and 3 kaolinite on average. Under the experiment conditions, the biotite can be regarded as nonreactive substance during short experiment duration.

### 2.2. Experimental Apparatus and Procedures

2.2.1. Experimental Apparatus

The experimental apparatus consisted of bioreactor, methane supply, temperature control system (air bath), and data acquisition system (Figure 1). The experiment was conducted in the low-temperature and high-pressure bioreactor, in which external control and monitoring of pressure and temperature were available with a volume of 1.0 L [23].

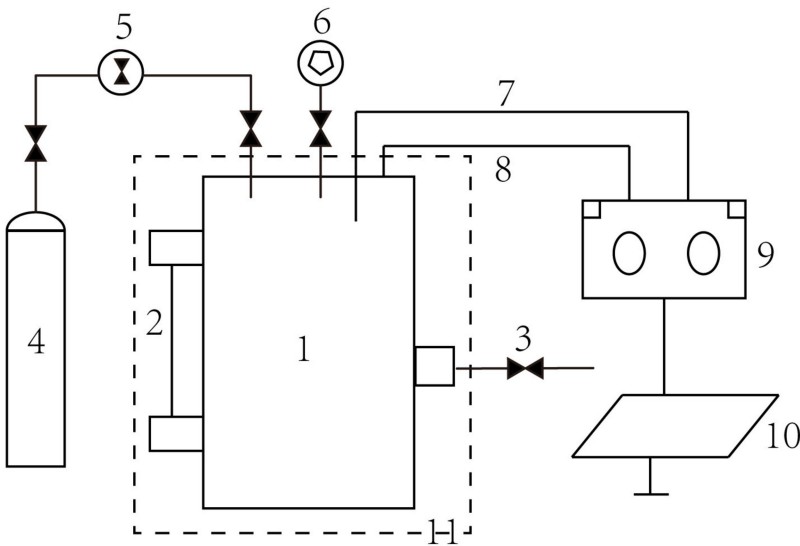

**Figure 1.** The experimental system. 1. Bioreactor, 2. Sight glass, 3. Sample collection, 4. Methane cylinder, 5. Regulator, 6. Vacuum Pump, 7. Temperature sensor, 8. Pressure sensor, 9. Data acquisition, 10. Computer, 11. Air bath.

2.2.2. Experimental Procedures

The marine sediments [24–26] cover a wide range of water depth, from about 470 m to 3000 m corresponding to the pressure from 4.7 MPa to 30 MPa and the range of temperature from 2.2 to 10 °C [27]. Microbial metabolism was extremely slow if the temperature below 4 °C. Previous experiment studies showed that when the temperature was higher than 4 °C, the metabolism of anaerobic microbes accelerated, and authigenic minerals are easier to form [19]. Therefore, the experimental temperature should be 10 °C. The experimental pressure, selected 6 MPa, was slightly lower than the critical equilibrium pressure at 10 °C for avoiding gas hydrate formation [28].

The amounts of 150 cm$^3$ sediments, 670 mL artificial pore water, 50 mL bacteria solution were put into the bioreactor in sequence. Sediment was at the bottom of the bioreactor, pore space of sediment was full of artificial pore water, and the surface of the water is 11 cm higher than that of sediment. Then, two pieces of granite slices were placed on the sediment surface and the bioreactor was quickly sealed. After evacuation, methane was injected into the bioreactor to rise and maintain pressure at the top of bioreactor.

Temperature (10 °C) and pressure (6 MPa) in the bioreactor were kept constant during the experiment. The experiment lasted 17 days, fluid samples were taken for analysis at regular intervals (1–2 days at the initial stage, 2–3 days in the late stage). Scanning electron microscope (SEM) and energy-dispersive X-ray spectrometer (EDS) were performed to view and analyze granite slices surface. 16S rDNA sequencing was used to determine bacteria diversity. More details about experimental procedures and methodology can be found in [19,20].

### 2.2.3. Analytical Methods

The Eh, optical density (OD), and the concentration of $HCO_3^-$, $Ca^{2+}$ of the fluid samples were measured. A batch of clean and evacuated sampling tubes were used to collect fluid samples through sample collection on the side of the bioreactor. These sampling tubes were sealed until measuring. $HCO_3^-$ concentration was determined following the acid-base neutralization titration method. Phenolphthalein, Methyl orange, and diluted hydrochloric acid were used in this method. $Ca^{2+}$ concentration was quantified by EDTA and an indicator of calconcarboxylic acid. OD was the absorbance of a material for a given wavelength ($\lambda = 600$) per unit distance and was measured with a UV-VIS spectrophotometer (T6 series, Purkinje General Corporation, Beijing, China). Eh was determined using HANNA multi-parameter water quality portable meter (HI9128, Woonsocket, RI, USA ). Instrument model of SEM, EDS, and XRD was JSM-6700F (JEOL Corporation, Tokyo, Japan), INCAX-SIGHT (Oxford, UK), and DX-2700 (Kemait NDT Co., Ltd., China), respectively. The accelerating voltage and working distance of the SEM were 15.0 kV and about 15 mm. All mineral morphologies were viewed using the backscattered electron (BSE) mode. The granite slices were detected using SEM with thin section form, and it was coated with gold dust before the test. The 16S rDNA sequencing was performed by Sangon Biotech (Shanghai, China).

## 3. Results

### 3.1. Fluid Chemistry and Microbes

The relative abundance of bacteria strain at the beginning and at the end of the experiment are shown in Figure 2. At the beginning, Citrobacter and other bacteria account for 86% relative abundance in the whole microbial community. The type of microbial community changes a lot at the end of the experiment. Pseudomonas account for 93% relative abundance at the end of the experiment and many bacteria at the beginning disappeared. The environment condition during the experiment have a great influence on microbial community, and it is beneficial to the growth of Pseudomonas. Figure 3a reflect the change of microbes concentration. The fluctuation in Figure 3a caused by microbial community replacement and mineral dissolution, which will be discussed in Section 4.1.

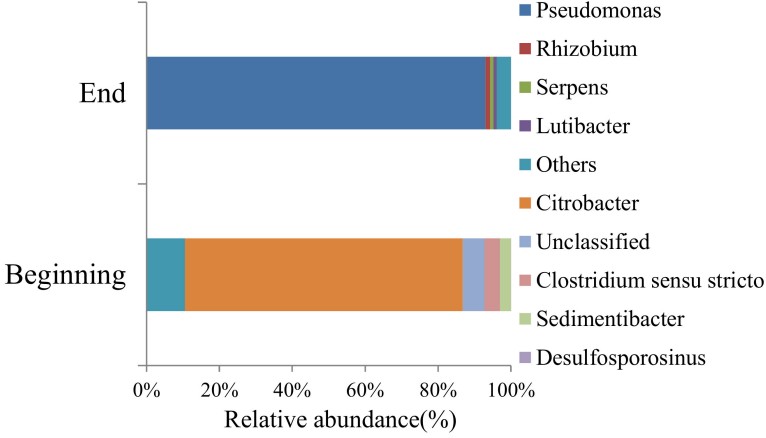

**Figure 2.** The microbes strain at the beginning and at the end of the experiment.

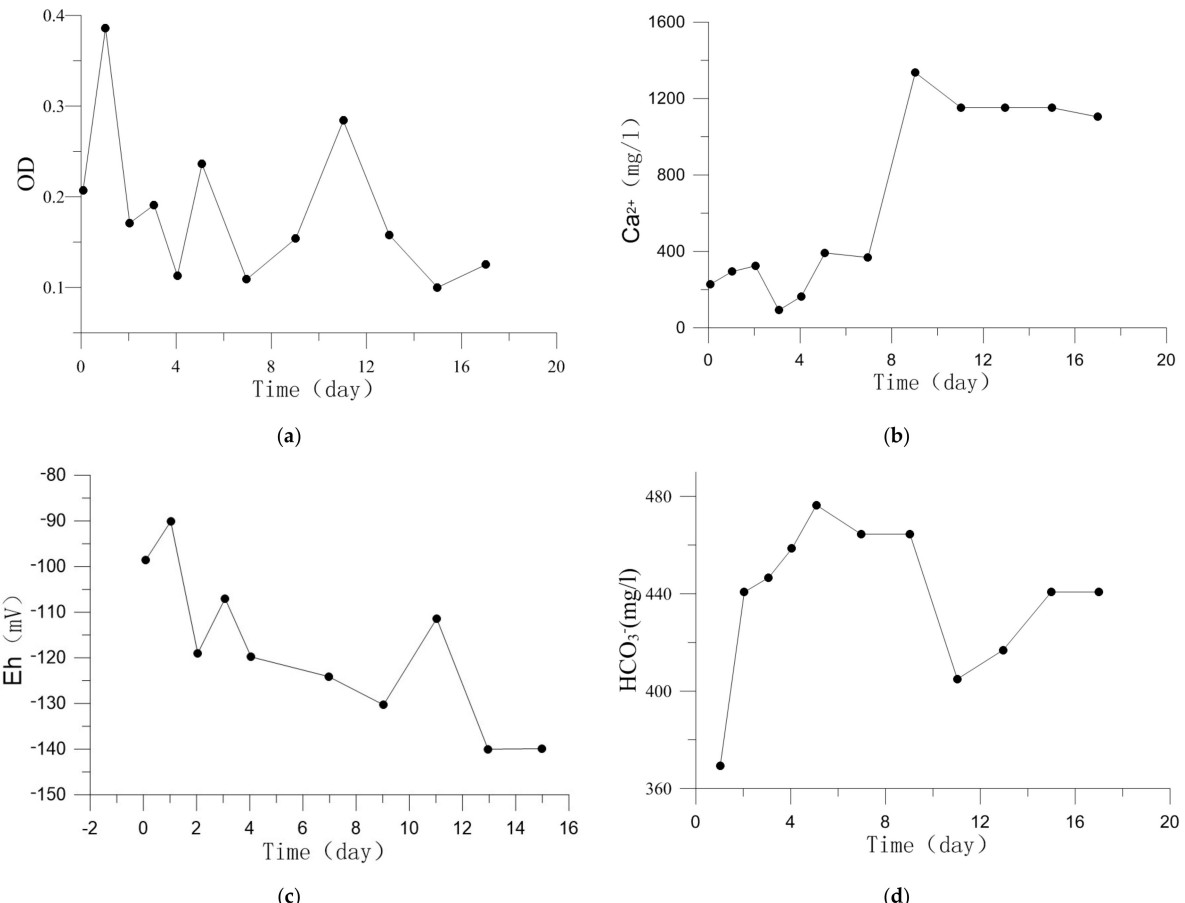

**Figure 3.** Fluid chemistry change during the experiment. (**a**) OD. (**b**) $Ca^{2+}$ concentrations. (**c**) Eh. (**d**) $HCO_3^-$ concentrations.

The evolution of $Ca^{2+}$ concentration is shown in Figure 3b. The dissolution of smectite-Ca contributed to its rise and the precipitation of carbonate make it less. More details were discussed in Section 4.1.

According to Equations (1)–(4), AOM produces $HCO_3^-$ and carbonates precipitation consumes $HCO_3^-$, both of which affect $HCO_3^-$ concentration in Figure 3d [29].

$$HCO_3^- + Ca^{2+} \rightarrow CaCO_3 + H^+ \qquad (2)$$

$$HCO_3^- + Mg^{2+} \rightarrow MgCO_3 + H^+ \qquad (3)$$

$$HCO_3^- + Fe^{2+} \rightarrow FeCO_3 + H^+ \qquad (4)$$

Low Eh value ($<-100$ mV) is favorable for AOM performance. The lower Eh value is, the more favorable for AOM performance [30]. Sulphur-containing compounds produced by microbial metabolism accumulate in pore water, which will cause the decrease of Eh [31–35]. Figure 3c shows Eh value dropped from $-90$ mV to $-140$ mV, it is a beneficial range for the growth and metabolism of microbes related to AOM [34]. Previous researches find that when Eh is less than 770 mV [36], the occurrence form of Fe is $Fe^{2+}$, and $Fe^{3+}$ disappears. Therefore, $Fe^{2+}$ was inferred to be the main form in the experiments.

### 3.2. Authigenic Minerals

The microscopic morphology and atomic composition of granite slices surface before and after the experiment was determined by SEM and EDS.

Before the experiment, low-albite (Figure 4a), quartz (Figure 4b,d), and k-feldspar (Figure 4c) were found on the slices. These minerals were the component of granite. Carbonates were found on the slices after the experiment (Figure 5). The faces and edges

of the mineral crystals are complete (Figure 5c, green square and yellow circles), and the edges are sharp (Figure 5c, yellow circles), indicating that the mineral was formed in situ without weathering. In addition, these minerals contained C, Fe, Ca, and Mg elements that granite lacked. Therefore, the minerals in Figure 5 were the authigenic carbonates formed during the experiment. EDS analyses showed that these carbonates mainly contained the elements C, O, Fe, Ca, and Mg. The minerals in Figure 5 were the mixture of $FeCO_3$, $MgCO_3$, and $CaCO_3$. Carbonates A was 15 μm × 8 μm in size with granular aggregate shape, and carbonates B was 20 μm × 16 μm in size with dense massive aggregate shape. The distribution range of carbonates C is large and uneven, showing amorphous morphology with an area of 5576 μm$^2$. The formation of carbonates was the direct consequence of AOM and the original record of methane seepage and carbon migration in marine sediments [37,38]. The direct cause of carbonates formation is $HCO_3^-$ reacting with $Fe^{2+}$, $Ca^{2+}$, and $Mg^{2+}$ in pore water.

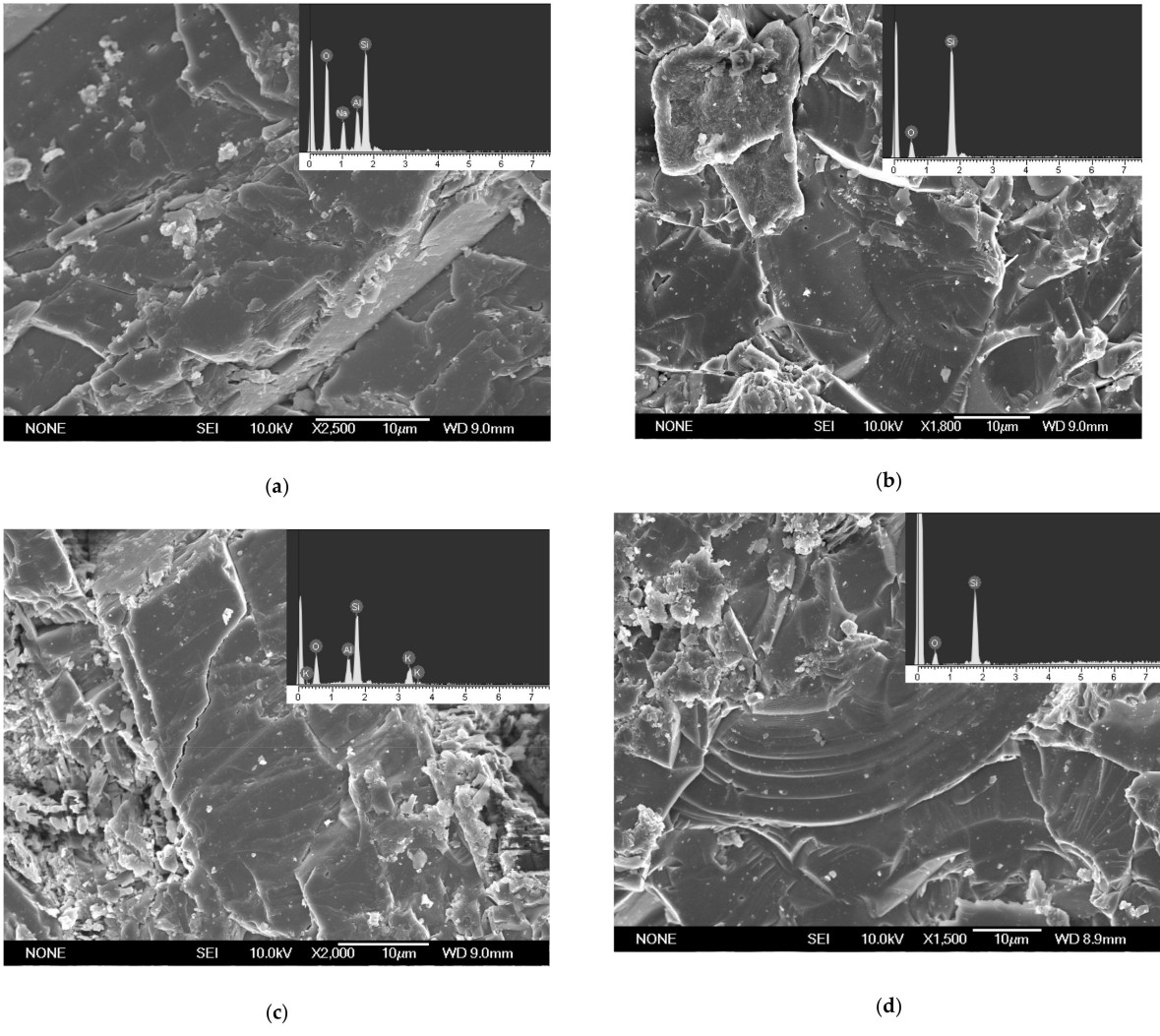

(a)

(b)

(c)

(d)

**Figure 4.** Granite slices surface prior to the experiment. (**a**) Low-albite. (**b**) Quartz. (**c**) K-feldspar. (**d**) Quartz.

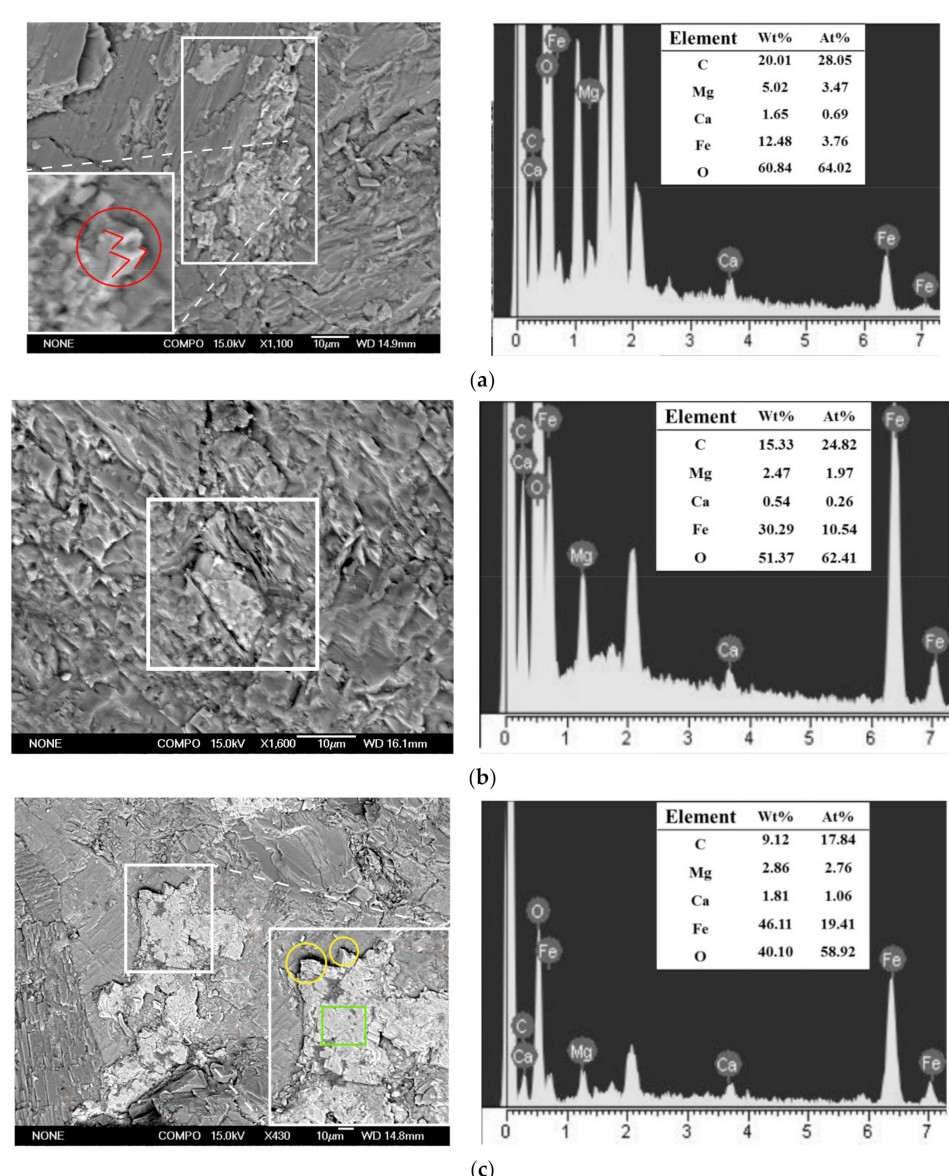

**Figure 5.** Authigenic carbonates formed after the experiment. (**a**) Carbonates A. (**b**) Carbonates B. (**c**) Carbonates C. Authigenic carbonates (white square); crystal faces (green square); crystal edges (yellow circle); rhombohedral twin (red circle); acute angle of rhombohedron (red line).

## 4. Discussion

### 4.1. The Role of Microbes in Chemical Composition Changes of Pore Water and Carbonates Formation

Citrobacter and pseudomonas were the dominant strain at the beginning and end of the experiment, respectively. The function of formation of carbonates induced by Citrobacter was reported by previous literature [18]. Ma et al. [39] injected Citrobacter into a bottle containing liquid medium. After 55 days' experiment, they found that a large amount of authigenic carbonates were formed in the bottle, which proved that Citrobacter has the biological function of promoting carbonates formation. The extracellular polymeric substances secreted by microbes contain polysaccharides and small molecular organic acids, which have the ability to adsorb and accumulate cations. Some work found that pseudomonas has the ability to secrete extracellular polymeric substances [40] and promote biological mineralization on carbonates [41–43].

The interaction between bacteria and minerals is an important process in terms of mineral formation. Clay minerals have the property of adsorbing bacteria, and some

bacteria have the ability to weather minerals. The sediments used in the experiment contained smectite, kaolinite, and illite. Smectite, kaolinite, and illite have the properties of adsorbing bacteria on the solid surface or in the soil solution [44–46]. The higher the concentration of electrolyte in the solution, the higher the adsorption rate of clay minerals to bacteria [44]. The concentration of electrolyte of pore water in the experiment is at least 710 mM. Therefore, we infer that the bacteria were adsorbed by clay minerals after the bacteria solution were put into the bioreactor.

Comparing the bacteria strain at the beginning and end of the experiment, we know that the amount of pseudomonas in the pore water increased during the experiment. Pseudomonas were capable of destroying the crystal structure of smectite, initiating it to dissolve [47]. The sediments used in the experiment were collected from shallow sea, so smectite-Ca is the main form of smectite. $Ca^{2+}$ and bacteria will be released into the pore water due to the dissolution of smectite-Ca [48]. The pseudomonas released destroyed the crystal structure of smectite-Ca, causing it to dissolve further. Therefore, $Ca^{2+}$ concentration in Figure 3b showed tendency to rise during the experiment and a mutation in 7–9 day. $Ca^{2+}$ concentration rose rapidly in 7–9 day increasing the ion product of carbonates and promotes the capture of $HCO_3^-$ in 9–11 day in Figure 3d.

### 4.2. The Formation Sequence of Carbonates of the Experiment

The carbonates formed in the experiment was dominant by $FeCO_3$, it was mixed with some $MgCO_3$ and $CaCO_3$. The isomorphism plays an important role in changing the composition of carbonates in the natural environment. Mixed components carbonates are usually formed from single component carbonate through isomorphism process [49]. From Table 2, we know the $FeCO_3$-$MgCO_3$ is complete isomorphous series. On the contrary, the $CaCO_3$-$FeCO_3$ and $CaCO_3$-$MgCO_3$ is incomplete isomorphous series or double salt. In the isomorphous series of $FeCO_3$-$MgCO_3$, Fe and Mg can be substituted with each other in any proportion. In the isomorphous series of $CaCO_3$-$FeCO_3$ and $CaCO_3$-$MgCO_3$, substitution between Fe, Ca, and Mg can only perform in a limited proportion range.

**Table 2.** Isomorphism among different carbonates.

| Isomorphous Series | Isomorphous Degree |
| --- | --- |
| $CaCO_3$-$MgCO_3$ | incomplete isomorphous series or double salt [49] |
| $CaCO_3$-$FeCO_3$ | incomplete isomorphous series [49] |
| $FeCO_3$-$MgCO_3$ | complete isomorphous series [49] |

The solubility product constant of $FeCO_3$ ($3.13 \times 10^{-11}$) is much smaller than that of $MgCO_3$ ($6.82 \times 10^{-6}$) and $CaCO_3$ ($2.8 \times 10^{-9}$) and $FeCO_3$ is the main carbonates formed in the experiment. We propose an inference that $FeCO_3$ is formed first in the experiment, and then the Fe in $FeCO_3$ is constantly substituted by Mg to form $FeCO_3$-$MgCO_3$ isomorphous series. Due to the large difference in cation radius, only incomplete isomorphous series and double salt can be formed between $CaCO_3$-$FeCO_3$ and $CaCO_3$-$MgCO_3$, so Ca accounts for a small proportion in the isomorphous mixture of carbonates. We will explain this inference in terms of atomic composition and morphology of carbonates in the following paragraphs.

In terms of atomic composition, the earlier $FeCO_3$ forms, the lower the ratio of Fe/Mg in $FeCO_3$-$MgCO_3$ isomorphous series. Within some siderite concretions in marine sediments of England, the Fe/Mg ratio decreases systematically from center to edge of concretions [50]. It is suggested that Fe can be continuously substituted by Mg in siderite in natural marine sediments. Therefore, we believe the carbonates C forms the latest (At% Fe/Mg = 7), followed by the carbonates B (At% Fe/Mg = 5), and the carbonates A formed the earliest (At% Fe/Mg = 1).

The morphology of mineral crystals and mineral aggregates reflects the formation time of minerals and external environmental conditions. The formation of mineral crystals needs a period of time. If there is enough time for minerals to precipitate, mineral crystals are easy to form characteristic morphology, otherwise, amorphous morphology is formed. Car-

bonates A shows granular aggregate morphology and rhombohedral twins (Figure 5a, red circle) with complete crystal structure. The rhombohedral twins are composed of two rhombohedral monocrystal with different orientations. The crystal of rhombohedral twins grows fully with complete crystal form and the formation time is earlier [49]. Rhombohedron or rhombohedral twins crystal is a typical crystal habit of carbonates. When the temperature decreases, the crystal form of carbonates transforms from obtuse rhombohedron to acute rhombohedron [51]. The low temperature set in the experiment has a good corresponding relationship with the formation of acute rhombohedron (Figure 5a, red circle and red line). The crystals of rhombohedron or rhombohedral twins and the granular aggregate or dense massive aggregate are typical characteristic morphology of $FeCO_3$-$MgCO_3$ isomorphous series [52]. Therefore, carbonates A is $FeCO_3$-$MgCO_3$ isomorphous series and it have been formed for a relatively long time. The morphology of carbonates B show dense massive aggregate. Carbonate B has formed a characteristic aggregate morphology, but the characteristic crystal form of $FeCO_3$-$MgCO_3$ isomorphous series has not been observed. This indicates that it formed later than carbonates A. Carbonates C shows irregular morphology, the crystal form of it is not developed. There is no characteristic morphology in crystal or in aggregate, indicating that the formation duration of carbonates C is very short and formed after carbonates A and carbonates B.

In conclusion, the sequence of carbonates formation in the experiment is carbonates A, carbonates B, carbonates C. We proves our inference in terms of atomic composition and morphology of carbonates.

### 4.3. Formation Stages of Authigenic Carbonates Induced by Microbes in the Marine Sediment

Authigenic minerals is the minerals formed in the ocean in situ, and authigenic carbonate is one of them. Authigenic carbonate is widely distributed in the modern ocean. Its composition and morphology are varied and was significantly affected by microbial action. However, the stage and model of authigenic carbonate formation induced by microbes in marine sediments have not been proposed. We will discuss this problem in this section. There are many ways for microbes to induce the formation of authigenic carbonates [53]. In this section, we only discuss the stage and model of authigenic carbonate formation performed by AOM and related microbes.

In order to distinguish the types of marine authigenic minerals, Zhu summarized the basis and contents of previous studies on the classification of the formation stages of marine authigenic minerals [49]. He proposed that the formation stage of authigenic minerals has the characteristically physical and chemical environment. Therefore, the formation stage of authigenic minerals should be regarded as the basis of the classification of authigenic minerals. On this basis, three stages of authigenic mineral formation are proposed in his theory: sedimentary stage (also known as hydrogenic stage), halmyrolysis stage, and syndiagenesis stage.

According to our work and previous studies, we propose a formation process of authigenic carbonates performed by AOM and related microbes in marine sediments as Figure 6. The initial stage, the concentration of $HCO_3^-$ in the pore water increases continuously due to the performance of AOM and related microbes, which leads to the increase of alkalinity in pore water [54]. The hydrogenic stage, according to the relationship between the ion product and solubility product of various carbonate minerals, a kind of carbonate mineral is preferentially formed. The halmyrolysis stage, according to the isomorphous degree of carbonates of different isomorphous series, isomorphism occurs between the carbonate formed in hydrogenic stage and different kinds of metal cations which are abundant in pore water of marine sediments. The isomorphous series of carbonates with two members, three members, or more were formed.

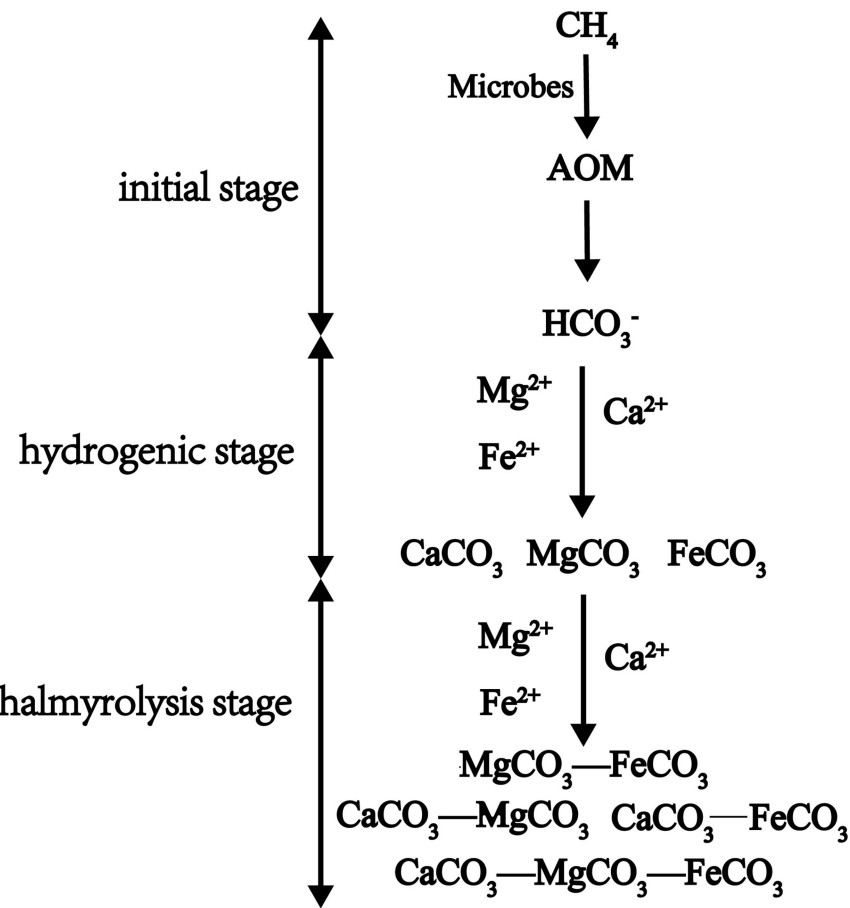

**Figure 6.** Formation stages of authigenic carbonates induced by AOM and related microbes.

The increase of alkalinity is a direct result of the performance of AOM, which has been confirmed in previous experiments and field observations [9,55,56]. It is the same trend of alkalinity increase in Figure 3d. The formation mechanism of single component carbonate is very clear and will not be discuss. The isomorphous series of carbonates with two members or three members was common in the field. The isomorphism of two members such as the high magnesium calcite, calcium magnesite, monohydrocalcite, aragonite [11,38,57], and Mg-doped amorphous calcium carbonate [58]. The isomorphism of three members ($CaCO_3$-$MgCO_3$-$FeCO_3$) such as siderite doped with Ca and Mg, ankerite in the field observation [38,59], mixture of $CaCO_3$-$MgCO_3$-$FeCO_3$ in the laboratory experiment [20,34].

The formation and evolution of carbonates in field are actually on long time scales, and isomorphous substitution of carbonates is complex with numerous members, such as Mn, Ni, Si, Zn, Pb, Sr, Ba, Co, etc. In this paper, we only consider the isomorphous series member of Ca, Mg, and Fe. The amount of authigenic carbonates formed in the experiment cannot reach the minimum amount limited by FTIR and XRD detection, so the mineral crystal structure and isomorphous series cannot be identified accurately, which needs to be further studied with longer experimental duration and better experimental design.

## 5. Conclusions

1. The microbial community changed during the experiment. Citrobacter and pseudomonas were the dominant strain at the beginning and end of the experiment, respectively. Both of them have the ability to promote the formation of carbonate minerals.
2. In the experiment, $FeCO_3$ is formed first, and then the complete isomorphous series of $FeCO_3$-$MgCO_3$ is formed. However, $CaCO_3$ accounts for a small proportion in the isomorphous mixture of carbonates.

3. We propose a formation process of authigenic carbonates performed by AOM and related microbes in marine sediments. It is divided into three stages. (1) The alkalinity of pore water increases during the initial stage. (2) A kind of carbonate mineral is preferentially formed during the hydrogenic stage. (3) The isomorphous series of carbonates were formed during the late halmyrolysis stage.

**Author Contributions:** Conceptualization, X.L.; investigation, J.W., Z.S., Z.W.; resources, S.S.; writing—original draft preparation, Z.W.; writing—review and editing, Y.C., H.T.; supervision, T.X. All authors have read and agreed to the published version of the manuscript.

**Funding:** This research was funded by National Natural Science Foundation of China, grant number 41877185.

**Institutional Review Board Statement:** Not applicable.

**Informed Consent Statement:** Not applicable

**Acknowledgments:** Thanks for Yukun Bai, Keqi Bei, Mingcong Wei, Fugang Wang, Chenghao Zhong for contributions and help in investigation and draft preparation.

**Conflicts of Interest:** The authors declare no conflict of interest. The funders had no role in the design of the study; in the collection, analyses, or interpretation of data; in the writing of the manuscript, or in the decision to publish the results.

## Appendix A

The medium composition was improved base on previous study [23] (g/L):$KH_2PO_4$ 0.744, $K_2HPO_4 \cdot 2H_2O$ 0.865, $MgCl_2 \cdot 6H_2O$ 11.896, $(NH_4)_2SO_4$ 0.2248, $CaCl_2 \cdot 2H_2O$ 1.3344, $Na_2SO_4$ 2.932, NaCl 23.3, pH = 7, $Na_2MoO_4 \cdot 2H_2O$ 0.02, $CuSO_4 \cdot 5H_2O$ 0.02, $FeSO_4 \cdot 7H_2O$ 0.01251, $ZnSO_4 \cdot 7H_2O$ 0.2, $MnCl_2 \cdot 4H_2O$ 0.00263, yeast extract 1, ascorbic acid 1.33, sodium lactate (wt. 50%) 10 mL/L, and demineralized water. Prior to the addition of the ascorbic acid and $FeSO_4 \cdot 7H_2O$, the medium was boiled in 121 °C 20 min. $H_2S$ is the metabolic product of AOM. In many incubation investigations, it has been proved that sulfides have an inhibiting effect on AOM performance and consortium growth [60,61]. In order to avoid the effect of sulfide on incubation, continuous culture mode and membrane bioreactors were used to perform incubation [62,63]. Therefore, $H_2S$ is not used in our microbes incubation.

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
