# Peer review of "Laboratory Experimental Study on the Formation of Authigenic Carbonates Induced by Microbes in Marine Sediments"

_jmse, doi:10.3390/jmse9050479_

Round 1

Reviewer 1 Report

This manuscript is a report on the laboratory experiment results of carbonate formation in conditions for anaerobic oxidation of methane. Generally, I welcome this contribution with great interest because little is known about the formation sequence of carbonates induced by AOM and about the transformation of physicochemical parameters of pore water due to this process. Nevertheless, the manuscript comes with some issues that I believe should be resolved before it is published in ‘Minerals’.

The lack of line numbering makes the form of comments difficult. I recommend to authors take this into account when submitting other articles.

  1. In the title manuscript, it is necessary to indicate that this is a study of a laboratory experiment;
  2. The manuscript would clearly benefit from presenting stable carbon isotope data of methane and precipitated carbonates. Moreover, stable oxygen isotope data of carbonates and water (before and after experiments) could also be added, if possible;
  3. In the last paragraph of section 4.1, the authors provide information on calcium sources and variability in calcium concentrations during the experiment. What about Fe and Mg? What is the Fe occurrence form?  How did the Fe and Mg concentrations change during the experiment?
  4. AOM is controlled by a consortium of methane-oxidizing archaea and sulfate reducing bacteria (Aharon and Fu, 2000; Borowski et al., 1996: Boetius et al., 2000; McGlynn et al., 2015 and etc.). Why didn't the autors use H2S for incubation of microbes strain? I think that this could have influenced the result of the experiment, because the iron sulfides would have formed and siderite could decrease.

Other issues:

In the first paragraph of the introduction, I propose to indicate which microbes control AOM (methane-oxidizing archaea and sulfate reducing bacteria).

The second paragraph of the introduction: «It was formed in different types such as calcite, siderite, aragonite, high-magnesium calcite, dolomite, ankerite in methane-bearing cold seeps of different areas» – also low-magnesium calcite is formed often.

The third paragraph of the introduction: «However, the particle size of carbonates in marine sediments is very small» - Authigenic carbonates  form rather large nodules often

The last paragraph of the introduction: « FeCO3 and FeCO3-MgCO3 complete isomorphous series were formed in sequence in the experiment» - This is the conclusion. I suggest removing it from the introduction.

Section 2.1.1. What is this smell? What it points to? Need to clarify or delete.

Section 2.1.1. «…met the requirements…» What are the requirements?

Section 2.2.3. In what form have carbonates been studied on SEM (thin section, polished section)? What material were the samples coated?

It is necessary to enlarge the photographs, improve the quality of the spectra and show the points from which the spectra were accumulated

Author Response

Comment 1: In the title manuscript, it is necessary to indicate that this is a study of a laboratory experiment;

Response 1:

Thanks for your suggestion, following your comment we changed the original title to “Laboratory Experimental Study on the Formation of Authigenic Carbonates Induced by Microbes in Marine Sediments” in this updated version of manuscript.

Comment 2: The manuscript would clearly benefit from presenting stable carbon isotope data of methane and precipitated carbonates. Moreover, stable oxygen isotope data of carbonates and water (before and after experiments) could also be added, if possible;

Response 2:

Thank you for your suggestion. We will conduct the isotopic testing in the future with a more perfect experimental design. Before the experiment, we consider to test the carbon isotopes of methane, water and carbonate. However, through communication with the testing agency, it is understood that carbonate should provide at least 2mg of samples to carry out isotope detection. As shown in Figure 5, the amount of carbonate produced in the experiment is too small, it is not easy to find it in granite slices surface when we performed SEM testing. Hence, isolation of sufficient quantities of carbonate for isotopic detection is not feasible. Therefore, the method of isotope was abandoned at that time. The isotopic results of water and methane were used for tracing carbonate source. If the carbonate samples are not made, the isotopes of water and methane will not be taken into account. Stable oxygen isotope is the same situation.

The testing agency is Beta Analytic, at the website: www.radiocarbon.cn/oxygen-isotopes/

Comment 3: In the last paragraph of section 4.1, the authors provide information on calcium sources and variability in calcium concentrations during the experiment. What about Fe and Mg? What is the Fe occurrence form?  How did the Fe and Mg concentrations change during the experiment?

Response 3:

The form of Fe was not determined in the experiments, the reason is that the total Fe and Fe2+ concentration in the experiment were not measured at the same time. But from Eh of the system we can inferred that Fe2+ was the main form of Fe. Eh in experiments carried out in this study ranged from -90mV to -140mV. Previous researches find that when Eh is less than 770mV [36], the occurrence form of Fe is Fe2+ and Fe3+ disappears. Therefore, Fe2+ was inferred to be the main form in our experiments. We determine total hardness of experimental solution in the last few days. we find that the Eriochome black T (indicator of total hardness) used in titration was far beyond the applicable amount. So we did not get the data of Mg concentration.

[36] Shen, z.e. Fundamentals of hydrogeochemistry. 1993.

Comment 4: AOM is controlled by a consortium of methane-oxidizing archaea and sulfate reducing bacteria (Aharon and Fu, 2000; Borowski et al., 1996: Boetius et al., 2000; McGlynn et al., 2015 and etc.). Why didn't the autors use H2S for incubation of microbes strain? I think that this could have influenced the result of the experiment, because the iron sulfides would have formed and siderite could decrease.

Response 4:

H2S is the metabolic product of AOM. In many incubation investigation, it has been proved that sulphide have an inhibiting effect on AOM performance and consortium growth [60] [61]. In order to avoid the effect of sulfide on incubation, continuous culture mode and membrane bioreactors was used to perform incubation [62][63]. Therefore, H2S is not benefit for microbes incubation.

[60] Zhang, Y., Henriet, J.P. Stimulation of in vitro anaerobic oxidation of methane rate in a continuous high-pressure bioreactor. Bioresource Technology 2010(101):3132-3138. doi:10.1016/j.biortech.2009.11.103.

[61] Chang, I.S., Clech, P.L. Membrane fouling in membrane bioreactors for wastewater treatment. Journal of Environmental Engineering 2002(128):1018-1029. doi:10.1061/(ASCE)0733-9372(2002)128:11(1018).

[62] Timmers, P.H.A., Gieteling, J. Growth of anaerobic methane-oxidizing archaea and sulfate-reducing bacteria in a high-pressure membrane capsule bioreactor. Applied and Environmental Microbiology 2015(81):1286-1296. doi:10.1128/AEM.03255-14.

[63] Girguis, P.R., Orphan, V.J. Growth and methane oxidation rates of anaerobic methanotrophic archaea in a continuous-flow bioreactor. Applied and Environmental Microbiology 2003(69):5472-5482. doi:10.1128/AEM.69.9.5472-5482.2003.

Other issues:

(1) In the first paragraph of the introduction, I propose to indicate which microbes control AOM (methane-oxidizing archaea and sulfate reducing bacteria).

Response: “Previous work estimated that more than 80% of the methane migrating upward through marine sediments was consumed through anaerobic oxidation of methane coupled with sulphate reduction (AOM) mediated by the microbial consortium” was changed to “Previous work estimated that more than 80% of the methane migrating upward through marine sediments was consumed through anaerobic oxidation of methane coupled with sulphate reduction (AOM) mediated by the microbial consortium of anaerobic methane-oxidizing archaea and sulfate-reducing bacteria.”

(2) The second paragraph of the introduction: «It was formed in different types such as calcite, siderite, aragonite, high-magnesium calcite, dolomite, ankerite in methane-bearing cold seeps of different areas» – also low-magnesium calcite is formed often.

Response:“It was formed in different types such as calcite, siderite, aragonite, high-magnesium calcite, dolomite, ankerite in methane-bearing cold seeps of different areas”was change to “It was formed in different types such as calcite, siderite, aragonite, high-magnesium calcite, low-magnesium calcite, dolomite, ankerite in methane-bearing cold seeps of different areas”and the supporting literature of low-magnesium calcite is [10].

[10] De Boever, E., Swennen, R. Lower Eocene carbonate cemented chimneys (Varna, NE Bulgaria): Formation mechanisms and the (a)biological mediation of chimney growth? Sedimentary Geology 2006(185):159-173. doi:10.1016/j.sedgeo.2005.12.010.

(3) The third paragraph of the introduction: «However, the particle size of carbonates in marine sediments is very small» - Authigenic carbonates form rather large nodules often.

Response: Large nodules is the aggregate morphology (large-scale) of carbonates, Here we are referring specifically to the small-scale crystal form. So we write “the particle size of carbonates”rather than “the morphology of carbonates”in this sentence.

(4) The last paragraph of the introduction: « FeCO3 and FeCO3-MgCO3 complete isomorphous series were formed in sequence in the experiment» - This is the conclusion. I suggest removing it from the introduction.

Response: Thanks, we removed this sentence from the introduction.

(5) Section 2.1.1. What is this smell? What it points to? Need to clarify or delete.

Response: Thanks for your comment. In the revised version of manuscript, it was deleted.

(6) Section 2.1.1. «…met the requirements…» What are the requirements?

Response: This sentence was changed to“Repeating this process until the microbial concentration and purity met the requirements (OD value of microbial solution reach 2.1), an enriched microbial solution was obtained.”

(7) Section 2.2.3. In what form have carbonates been studied on SEM (thin section, polished section)? What material were the samples coated?

Response: In the revised manuscript, after“All mineral morphologies were viewed using the backscattered electron (BSE) mode.” we added a sentence “The granite slices was detected using SEM with thin section form and it was coated with gold dust before test.”

(8) It is necessary to enlarge the photographs, improve the quality of the spectra and show the points from which the spectra were accumulated.

Response: A clearer picture with higher dpi and non-compressed style has been used in the revised manuscript.

Reviewer 2 Report

The work is very well organized also the aims of the authors, who try to study the formation process of authigenic carbonate in more depth than the previous studies that are limited to determine the only total amount of minerals.
The testing methodologies are also clear and detailed.
The content of this paper meets the aims of this Journal and I agree with its publication. However, the authors should improve the English language and review my few requests, which are outlined below:

1. Some parts of the text between the abstract and the introduction are the same, I suggest finding different words.

2. In the section "Artificial pore water" we talk about seawater of Qiongdongnan basin (South China Sea). The caption indicates "The major component of artificial pore water".
 Was the water used from this basin or artificial and made in the laboratory following the species concentration specifications in table 1?
Are only these species important to know or should a more thorough characterization be used?

3. Are the experimental procedures standard or is it a methodology already used by you or found in the current literature?

4. Describe Figure 3 more accurately, referring to the letters used a, b, c, and d.

Author Response

Comment 1: Some parts of the text between the abstract and the introduction are the same, I suggest finding different words.

Response 1:

“This work provide experimental experience and reference basis for further understanding the formation mechanism of authigenic carbonates in marine sediments. ” in the abstract part appeared again in the introduction part. We changed this sentence in the introduction to“This work improves our understanding of the formation of authigenic carbonates from a relatively microscopic perspective”

Comment 2: In the section "Artificial pore water" we talk about seawater of Qiongdongnan basin (South China Sea). The caption indicates "The major component of artificial pore water". Was the water used from this basin or artificial and made in the laboratory following the species concentration specifications in table 1? Are only these species important to know or should a more thorough characterization be used?

Response 2:

The artificial pore water used in this study is water made in the laboratory. The composition of artificial water was same with that of pore water in Site T1 in Qiongdongnan basin (South China Sea). The composition of pore water in Site T1 was characterized using component of pore water in marine sediment. Therefore, it is reasonable to use this composition. The details can be found in section 2.1.2. And the caption of Table 1 was changed to “The composition of artificial pore water”.

Comment 3: Are the experimental procedures standard or is it a methodology already used by you or found in the current literature?

Response 3:

The experimental procedures and methodology was already used by previous researchers (e.g., [19] [20]), and more details can be found in these literatures.

[19]Wei, m. The formation of authigenic minerals during methane seeping in seafloor: insight from laboratory test; Jilin university, 2016.

[20]      Xu, T.F., Bei, K.Q. Laboratory experiment and numerical simulation on authigenic mineral formation induced by seabed methane seeps. Marine and Petroleum Geology 2017(88):950-960. doi:10.1016/j.marpetgeo.2017.09.025.

Comment 4: Describe Figure 3 more accurately, referring to the letters used a, b, c, and d.

Response 4:

Thanks for your suggestion, the description of Figure 3 in the manuscript was changed in the form of “Fig3.b show that or Fig3.a indicate” rather than the form of “The concentration of Ca2+ or OD value change”.

Round 2

Reviewer 1 Report

All my comments have been corrected. The manuscript can be accepted.

Author Response

Thank you for your kind suggestion.